# The importance of snow albedo for ice sheet evolution over the last glacial cycle

Matteo Willeit and Andrey Ganopolski

Potsdam Institute for Climate Impact Research, Potsdam, Germany

**Abstract.** The surface energy and mass balance of ice sheets strongly depends on the amount of solar radiation absorbed at the surface, which is mainly controlled by the albedo of snow and ice. Here, using an Earth system model of intermediate complexity, we explore the role played by surface albedo for the simulation of glacial cycles. We show that the evolution of the Northern Hemisphere ice sheets over the last glacial cycle is very sensitive to the representation of snow albedo in the model. It is well known that the albedo of snow depends strongly on snow grain size and the content of light-absorbing impurities. Excluding either the snow aging effect or the dust darkening effect on snow albedo leads to an excessive ice build-up during glacial times and consequently to a failure in simulating deglaciation. While the effect of snow grain growth on snow albedo is well constrained, the albedo reduction due to the presence of dust in snow is much more uncertain, because the light-absorbing properties of dust vary widely as a function of dust mineral composition. We also show that assuming slightly different optical properties of dust leads to very different ice sheet and climate evolutions in the model. Conversely, ice sheet evolution is less sensitive to the choice of ice albedo in the model. We conclude that a proper representation of snow albedo is a fundamental prerequisite for a successful simulation of glacial cycles.

## 1. Introduction

The net surface mass balance of ice sheets is equal to the difference between accumulation, which is controlled by the hydrological cycle, and of ablation, which is determined by the surface energy balance. The surface energy balance strongly depends on the amount of solar radiation absorbed at the surface. While the amount of radiation reaching the surface is mainly determined by the insolation at the top of the atmosphere and cloud cover, the fraction of radiation absorbed at the surface is controlled by its albedo. Since ice sheets are mostly covered by snow, the albedo of snow plays a crucial role for the surface energy and mass balance of ice sheets.

The importance of snow and ice albedo parameterizations for ice sheet modeling has been known for long time. To the contrary, the role of snow albedo parameterization on modeling of glacial cycles is much less understood. Most of previous simulations of glacial cycle(s), e.g. Bonelli et al. (2009), Tarasov and Richard Peltier (2002), Zweck and Huybrechts (2005), Charbit et al. (2007), Abe-Ouchi et al. (2007), Lunt et al. (2008), Gregoire et al. (2012), Liakka et al. (2016) and many others employed the so-called positive degree day (PDD) scheme, which does not account explicitly for snow and ice albedoes.

The albedo of snow is a complex function of snow grain size and concentration of light-absorbing impurities (Warren, 1982; Warren and Wiscombe, 1980). After snowfall, snow crystals undergo rapid transformations in size and shape, with a tendency for snow grains to grow larger with time (Colbeck, 1982). The rate of change is controlled by snow temperature and the temperature gradient inside the snow, with melt-freeze cycles beeing additionally very efficient in accelerating grain growth during snowmelt (Brun et al., 1992; Flanner and Zender, 2006). The change in snow grain size affects the interaction of the snow surface with the incoming solar radiation, with larger grains increasing the path that photons are traveling in the snow and therefore decreasing its albedo. The decrease in albedo due to a growth of the optically equivalent snow grain size from 100 μm, typical for fresh snow, to 1000 μm, typical for melting snow, is ~10 % (Fig. 1).

Both radiative transfer models (Aoki et al., 2011; Hadley and Kirchstetter, 2012; Warren and Wiscombe, 1980) and direct measurements (Bryant et al., 2013; Doherty et al., 2013; Gautam et al., 2013; Painter et al., 2010, 2012; Skiles et al., 2012; Skiles and Painter, 2017) demonstrate that even small amounts of light absorbing impurities (LAI) affect the surface albedo of snow significantly. Black carbon and desert dust are the main sources of LAI in snow, but algal blooms and organic carbon could also play a role. Black carbon concentrations of less than 1 ppmw (parts per million in weight) in fresh snow can already cause decreases in albedo by several percent (Warren and Wiscombe, 1980). The effect of mineral dust on snow albedo is much lower, with 1 ppmw of black carbon being equivalent to roughly 100-200 ppmw of mineral dust (Dang et al., 2015; Warren and Wiscombe, 1980). Algal blooms over Greenland have been shown to substantially reduce surface albedo (Lutz et al., 2016; Musilova et al., 2016; Stibal et al., 2017). Simulations show that changes in albedo due to LAI might significantly affect the surface mass balance of ice sheets during glacial times (Ganopolski et al., 2010; Krinner et al., 2006), at present (Dumont et al., 2014; Tedesco et al., 2016) and in future climate change scenarios (Goelles et al., 2015).

In this study we explore the sensitivity of ice sheet and climate evolution over the last glacial cycle to the representation of snow albedo in the CLIMBER-2 Earth system model of intermediate complexity. We limit the scope of our study to the effect of snow aging and mineral dust concentration in snow. Several lines of evidence suggest that dust deposition was substantially larger during glacial times (Kohfeld and Harrison, 2001; Lambert et al., 2015; Mahowald et al., 2006), particularly also at the southern margins of the NH ice sheets, the areas most affected by ablation. Dust is therefore likely to be an important player for the ice sheet ablation through its effect on snow albedo. The effect of black carbon is neglected in this study. Although it has been suggested that the effect of black carbon on surface albedo might play an important role in the present day climate (Flanner et al., 2007; Hansen and Nazarenko, 2004; Yasunari et al., 2015), most of the black carbon which is deposited over the boreal region comes from sources related to industrial activities (Bauer et al., 2013; Lamarque et al., 2010). We therefore assume that black carbon deposition over regions potentially covered by ice sheets was negligible in pre-industrial times. The effect of other LAI, like algae, are still far from being properly understood and are therefore not considered in the present study.

It is now becoming possible to use complex Earth system models based on general circulation models to simulate the last glacial cycle (e.g. Latif et al., (2016)). In these models, over-simplistic obsolete schemes like PDD will be substituted by a physically based energy balance approach similar to that used in CLIMBER-2. An exploration of the impact of albedo parameterization on glacial cycles simulations with a computationally efficient model like CLIMBER-2 can provide useful insights into which processes are important and should therefore be accounted for.

## 2.    Methods

### 2.1.    The CLIMBER-2 model

CLIMBER-2 (Ganopolski et al., 2001; Petoukhov et al., 2000) includes a coarse resolution statistical-dynamical atmosphere model, a 3 basin zonally averaged ocean model, a land surface and vegetation model (Brovkin et al., 1997) and the 3-d polythermal ice sheet model SICOPOLIS (Greve, 1997). SICOPOLIS is applied only to the Northern Hemisphere, with a resolution of 1.5° x 0.75°. Antarctica is prescribed from present day observations. The climate component and SICOPOLIS are coupled once per 10 years through a surface energy and mass balance interface module (SEMI) (Calov et al., 2005). SEMI performes a physically based 3-dimensional downscaling of climatological fields from the coarse atmospheric grid to the ice sheet model grid and computes the surface mass balance and the surface temperature using a physically based surface energy balance approach.  Importantly, precipitation is downscaled accounting for the slope effect and the desert-elevation effect. Radiation and atmospheric temperature and humidity are first interpolated bilinearly and then corrected for the

surface elevation of the ice sheet. Refreezing is accounted for as a constant fraction, 0.3, of surface melt. Computed annual fields of surface ice sheet mass balance and of surface temperature are used in SICOPOLIS as surface boundary conditions. In turn, SICOPOLIS feeds back the average ice sheet elevation, the fraction of land area covered by ice sheets, the sea level and the freshwater flux into the ocean from the ablation of ice sheets and from ice calving to the climate component.

The model has been used to explore the hysteresis in the climate-cryosphere system (Calov and Ganopolski, 2005) and has successfully simulated the last eight glacial cycles (Ganopolski et al., 2010; Ganopolski and Calov, 2011). It has been used to explore the effect of dust radiative forcing on glacial-interglacial cycles (Bauer and Ganopolski, 2014), the impact of permafrost on simulation of glacial cycles (Willeit and Ganopolski, 2015) and the initiation of Northern Hemisphere glaciation (Willeit et al., 2015).

The model version used in this study includes a fully interactive dust cycle as described in (Bauer and Ganopolski, 2010, 2014). The direct radiative forcing of dust loading in the atmosphere is explicitly accounted for and dust deposition at the surface affects snow albedo both in the land surface module and in SEMI. Compared to (Bauer and Ganopolski, 2010) we replaced the precipitation dependence of dust emissions with a relative soil moisture ($\theta$) dependence, so that their equation (12) for the threshold value for the climatological wind speed for dust emissions becomes:

$$u_t = u_0\big(1 + \tanh\big(c_\theta(\theta - \theta_t)\big)\big),$$

where $u_0 = 3$ m/s is the reference threshold wind speed, $\theta_t = 0.3$ is the soil moisture of transition from semi-arid to humid conditions, and $c_\theta = 10$ is a normalization constant.

We use the parameters corresponding to the solution L1 in Bauer and Ganopolski (2014), which assumes that the fraction of precipitation-driven wet dust deposition is 70 % of the total, and an imaginary refractive index of airborne dust of 0.003.

The dust deposition on ice sheets further includes dust from simulated sediments produced by glacial erosion. This dust
source is not included in the global dust cycle model due to its very local origin, which can not be represented on the coarse atmospheric grid. Dust deposition produced from glaciogenic sources is parameterized based on the assumption that the emission of glaciogenic dust is proportional to the delivery of glacial sediments to the edge of an ice sheet (see Ganopolski et al. 2010 Appendix A for details). Most of the glaciogenic dust originates from the southern flanks of the ice sheets and this source is significant only for mature ice sheets, which reach well into areas covered by thick terrestrial sediments.

**2.2.     Snow albedo parameterization**

Three components in the parameterization of snow albedo used in CLIMBER-2 are critically important, namely, the aging of pure snow, the concentration of light-absorbing impurities in snow from dust deposition and the synergy between aging of snow and impurities (Warren, 1982; Warren and Wiscombe, 1980). Under "synergy" we understand here the fact that the effect of impurities on snow albedo is much higher for "old" snow than for fresh snow. The parameterisations described
below are applied both to the surface scheme on the atmospheric grid and to SEMI on the ice sheet grid. The surface albedo over ice sheets is computed as:

$$\alpha = f_{snow}\alpha_{snow} + (1 - f_{snow})\alpha_{ice},$$

where $f_{snow}$ is the grid cell fraction which is considered to be snow covered, $\alpha_{snow}$ is the albedo of snow and $\alpha_{ice}$ is the albedo of bare ice. $\alpha_{ice}$ is set to 0.4. $f_{snow}$ is 1 if the snow water equivalent in the grid cell is larger than 30 kg/m$^2$ and linearly related to the ratio between snowfall and ablation if the snow water equivalent is below 30 kg/m$^2$. Snow albedo is
computed for two spectral bands (visible and near infrared radiation) and separately for direct beam and diffuse radiation. The diffuse albedo values are a function of snow grain size and dust concentration at the surface following Warren and Wiscombe (1980) (their Fig. 5 for dust radius of 1 μm and imaginary refractive index of 0.01). It is shown in Fig. 1.

A snow aging factor is used to represent the grain size evolution and its effect on albedo, similarly to Dickinson et al. (1986). The snow age factor, $f_{age}$, is parameterized as a function of snow temperature $T_s$ and snowfall rate $S$ on each atmospheric time step (one day) as:

$$f_{age} = 1 - \frac{\ln\left(1 + f_{age}^T \frac{S_c}{S}\right)}{f_{age}^T \frac{S_c}{S}},$$

with $S_c = 2 \cdot 10^{-5}\ \mathrm{kg\,m^{-2}s^{-1}}$ and

$$f_{age}^T = e^{0.05(T_s - T_0)} + e^{T_s - T_0}.$$

$$T_0 = 273.15\ K.$$

5    The snow grain size, $r_e$ in µm, and snow age factor are related by:

$$r_e = 50 + 200 \cdot \left(10^{f_{age} \cdot \log_{10}\left(1 + \frac{1000 - 50}{200}\right)} - 1\right).$$

The dust mass concentration in snow is simply computed as the ratio of dust deposition rate and precipitation rate. During snowmelt dust is assumed to concentrate near the snow surface and dust concentration is allowed to increase by up to a factor of five, consistent with observations for the top 4 cm presented in Doherty et al. (2013).

The direct beam snow albedo values depend on the solar zenith angle and the standard deviation of orography, $\sigma_z$, as:

$$\alpha_{snow}^{vis,dir} = \alpha_{snow}^{vis,dif} + f_{oro}f_{cosz}\left(1 - \alpha_{snow}^{vis,dif}\right),$$

10    where

$$f_{oro} = 0.4 \cdot \left(1 - \tanh\frac{\sigma_z}{1000}\right),$$

$$f_\mu = 0.5 \cdot \left(\frac{3}{1 + 2\mu} - 1\right),$$

and µ is the cosine of the solar zenith angle.

To test the sensitivity of our results to the representation of snow albedo, we have additionally introduced two alternative parameterisations of snow albedo. The first one is from Dang et al. (2015) and the second from Gardner and Sharp (2010). 15 Both include the effect of snow grain size and black carbon content. The effect of dust is computed through a black carbon equivalent following Dang et al. (2015). The two alternative parameterisations are compared to the standard one in Fig. 1. The different models agree on a ~10 % albedo reduction caused by the aging of snow, for a snow grain growth from ~100 to ~1000 µm (Fig. 1a). The impact of dust concentration on fresh snow albedo is generally larger but much more uncertain, ranging between 15-25 % albedo reduction for a dust concentration of 1000 ppmw relative to pure snow (Fig. 1c). The 20 differences can be largely explained by the choice of the imaginary refractive index of dust. The imaginary refractive index of dust varies over an order of magnitude as a function of dust composition (e.g. Fig. 7 in Dang et al. (2015)) and this range of possible values is reflected in the differences in albedo seen in Fig. 1. The combination of aged snow with high dust concentrations reduces snow albedo to values below 0.4 (Fig. 1b,d).

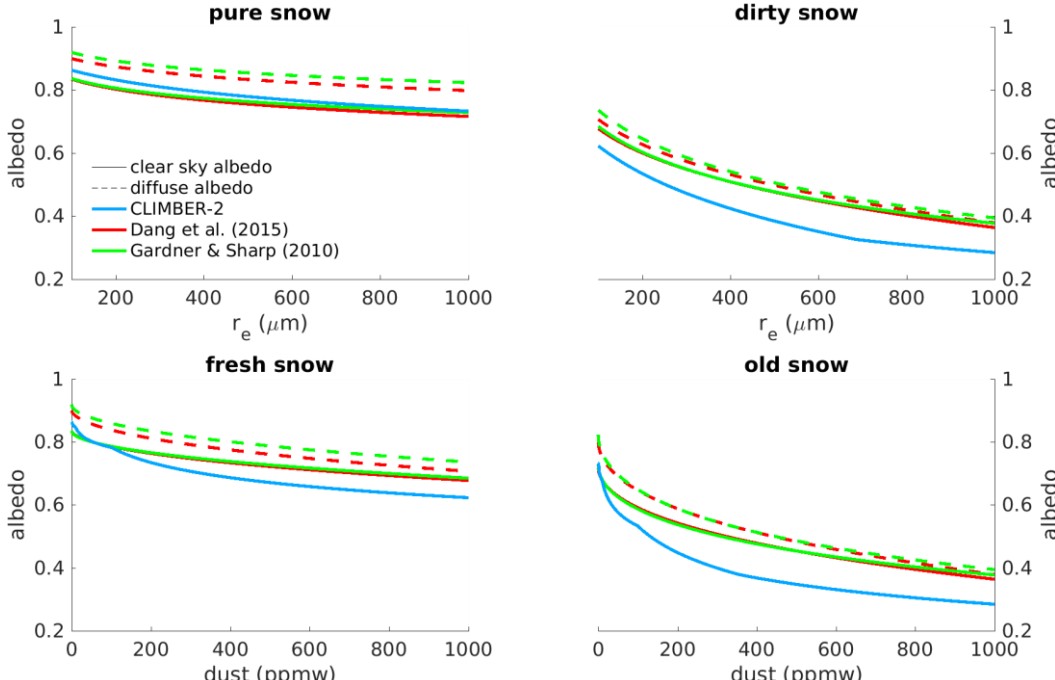

**Figure 1: Clear sky and diffuse snow albedo dependence for a number of different parameterisations (Dang et al., 2015; Gardner and Sharp, 2010; Warren and Wiscombe, 1980) as indicated in the legend. a) Pure snow albedo dependence on effective snow grain radius. b) Fresh snow albedo ($r_e = 100$ μm) as a function of dust concentration. c) Dirty snow albedo (dust concentration of 1000 ppmw) as a function of snow grain size. d) Old snow albedo ($r_e = 1000$ μm) as a function of dust concentration. The clear sky albedo is for a solar zenith angle of 50°. In CLIMBER-2 the snow albedo for diffuse and direct radiation is identical for solar zenith angles below 60°.**

## 2.3.    Experiments

We used CLIMBER-2 to simulate the last glacial cycle, from 130 ka (1000 years ago) to the present day. The transient model simulations are driven by orbital forcing (Laskar et al., 2004) and the time-varying concentration of greenhouse gases expressed as equivalent CO2 concentration (Ganopolski et al., 2010). The initial condition is the equilibrium climate state computed with greenhouse gas concentration and orbital forcing of the preindustrial period.

First we performed a reference model simulation using the standard CLIMBER-2 surface albedo parameterisation. Then we run a set of offline simulations in which, similarly to Bauer and Ganopolski (2017), the climate and ice sheets are prescribed from the reference simulation and the surface mass balance is diagnosed for experiments with different surface albedo setups. To separate the importance of snow aging and dust on snow albedo, we run offline experiments with and without snow aging and with and without the effect of aeolian and glaciogenic dust sources on snow albedo.

Finally we performed a set of online simulations using the different surface albedo setups but this time allowing bidirectional coupling between the climate and ice sheet models. Additional experiments with the alternative albedo parameterisations described in Section 2.2 are used to explore the sensitivity to different snow albedo schemes. We also tested the model sensitivity to different values of ice albedo, ranging from 0.3 to 0.5, and to different global dust emissions scaling factors (from 1/4x to 4x). All online experiments are listed in Table 1.

**Table 1: List of online model simulations.**

|       | Snow aging | Dust deposition   | Snow albedo | Ice albedo |
|-------|------------|-------------------|-------------|------------|
| REF   | On         | On                | Std         | 0.4        |
| A1D0  | On         | Off               | std         | 0.4        |
| A0D1  | Off        | On                | std         | 0.4        |
| A0D0  | Off        | Off               | std         | 0.4        |
| A1D1g | On         | Glaciogenic only  | std         | 0.4        |

| A1D1p1 | On | On | Gardner & Sharp | 0.4 |
|---|---|---|---|---|
| A1D1p2 | On | On | Dang et al | 0.4 |
| A1D1i03 | On | On | Std | 0.3 |
| A1D1i05 | On | On | Std | 0.5 |
| A1D1d1/4 | On | On, ¼ x | Std | 0.4 |
| A1D1d1/2 | On | On, ½ x | Std | 0.4 |
| A1D1d2 | On | On, 2 x | Std | 0.4 |
| A1D1d4 | On | On, 4 x | Std | 0.4 |

## 3.        Results and discussion

Figure 2 shows the evolution of several modelled variables over the last glacial cycle in the reference simulation. The global temperature decreases by ~ 6°C from the Eemian interglacial (126 ka) to the last glacial maximum (LGM, 21 ka) (Fig. 2a). The modelled sea level variations agree reasonably well with available reconstructions (Spratt and Lisiecki, 2016), with a minimum sea level ~120 m below the present day during the LGM (Fig. 2b). The largest contribution to sea level comes from the Laurentide ice sheet (Fig 2c). The surface mass balance of the NH ice sheets is positive through most of the last glacial cycle, except for the deglaciation period between 20 and 10 ka (Fig. 2d), when the ablation rate exceeds the accumulation rate (Fig. 2e,f).

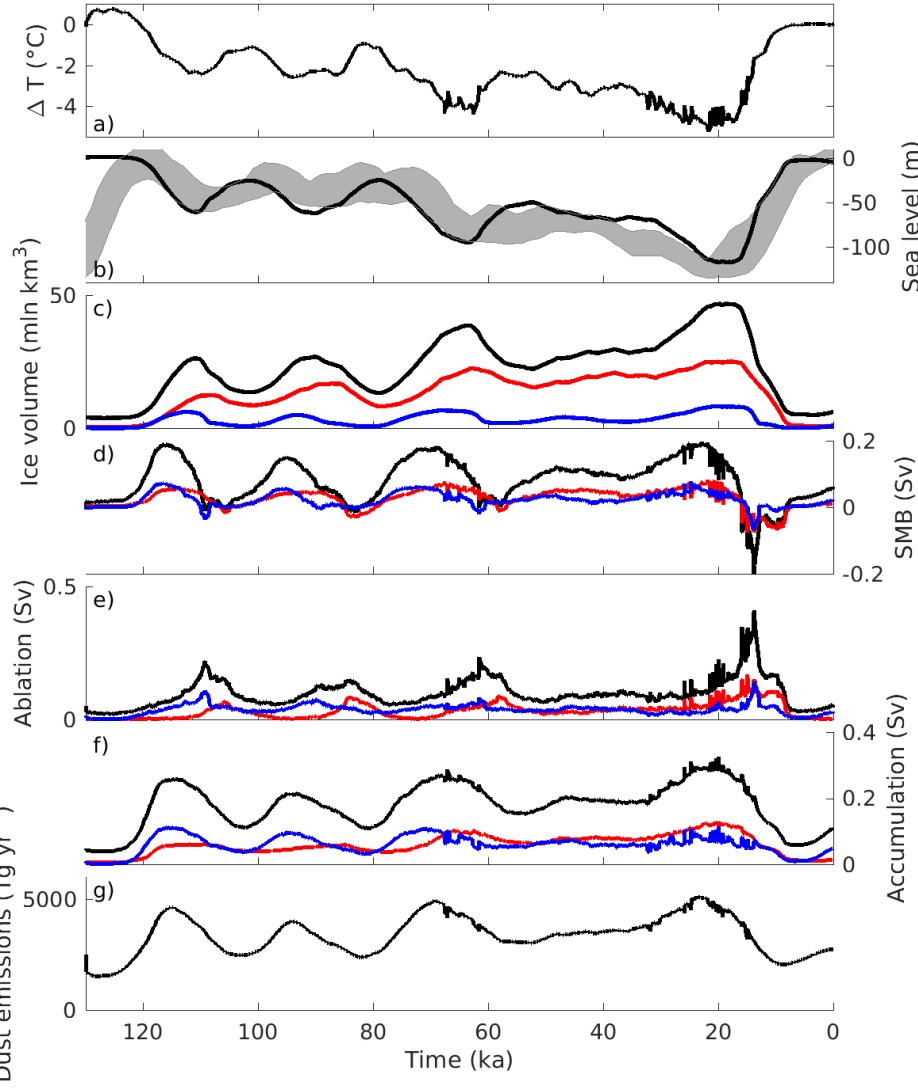

**Figure 2: Simulated a) global temperature anomaly, b) sea level, c) ice volume, d) surface mass balance, e) ablation, f) accumulation and g) global dust emissions for the reference experiment. The red and blue lines in c-f represent the Laurentide and Fennoscandian ice sheet, respectively. The shading in b) is the sea level range from Spratt and Lisiecki (2016).**

The reference simulation in this study differs from previous CLIMBER-2 last glacial cycle simulations presented in Bauer and Ganopolski (2014) and Ganopolski et al. (2010) in that it includes a fully interactive dust cycle, as described in Section 2.1. The modelled dust deposition for the present day and the LGM is compared to available observation-based estimates in Fig. 3. A detailed model evaluation based on these data is challenging, because of the large variability among different observation-based reconstructions (Fig. 3b,c and Fig. 3e,f) (Lambert et al., 2015; Mahowald et al., 1999, 2006). Even at present the estimated global value of dust deposition and atmospheric load varies largely between different studies (e.g. Table 1 in Bauer and Ganopolski (2014)). Ganopolski et al. (2010) used the Mahowald et al. (1999) data for present day and LGM, with their relative contributions scaled with sea level, as prescribed forcing in the surface energy and mass balance interface module. The climate-ice sheet model has therefore been tuned for dust amounts similar to Mahowald et al. (1999). Hence, to avoid the need to retune the model, we scaled the model dust emissions by adjusting the dimensionless global calibration constant $c_q$ in Eq. (8) of Bauer and Ganopolski (2010) to get a present-day global dust deposition of ~3000 Tg/yr (Fig. 2g), comparable to Mahowald et al. (1999).

. At LGM the global dust deposition is roughly doubled in our simulations (Fig. 2g). What is more important for the impact on surface albedo is the spatial distribution of dust deposition. In general, at present, the dust deposition pattern seen in observations is reasonably well captured by the model, although it tends to slightly overestimate the annual dust deposition at high northern latitudes (Fig. 3a-c). At the LGM, the modelled geographic distribution of dust deposition resembles in many aspects the reconstructions from Mahowald et al. (1999), with increased dust deposition over Siberia and at the southern boundary of the Laurentide ice sheet over North America (Fig. 3d,f). Although the LGM dust deposition pattern is similar also in the reconstructions of Lambert et al. (2015), the absolute values are substantially larger in the latter compared to the model or Mahowald et al. (1999).

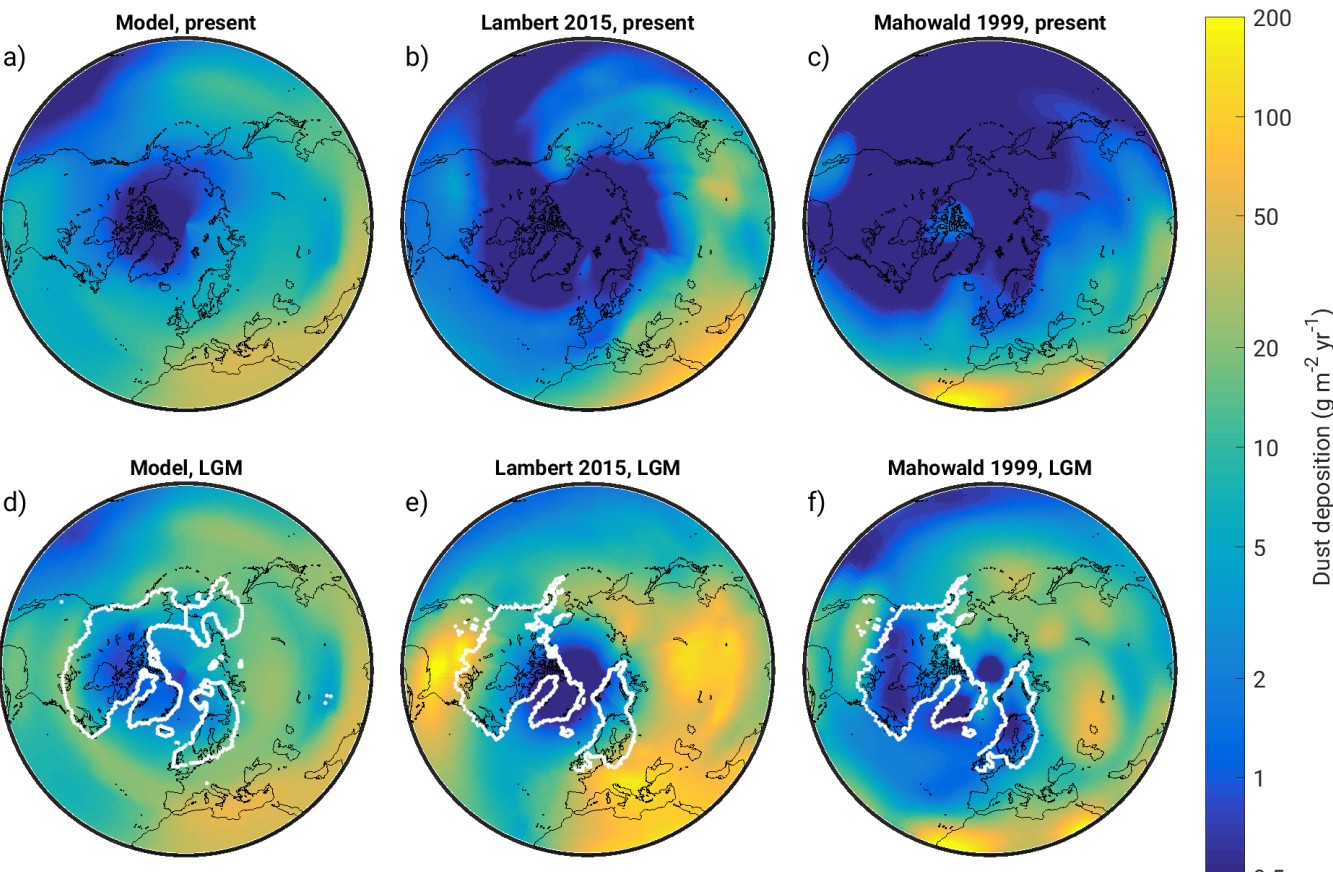

**Figure 3: Modelled annual dust deposition compared to observations for the present day (top) and reconstructions for the last glacial maximum (bottom). The model results (left) are compared to data from Lambert et al., 2015 (middle) and Mahowald et al.,**

**1999 (right). The dataset of Mahowald et al. 1999 does not include glaciogenic dust. For the LGM, the white lines indicate the modelled (d) and reconstructed (e,f, (Tarasov et al., 2012)) extent of the ice sheets.**

Due to the highly non-linear nature of the climate-cryosphere system, relatively small changes in the model can have a very large impact on the simulated coupled system response to orbital forcing. The offline simulations with prescribed climate and ice sheets from the reference simulation provide a mean to understand the impact of the different factors affecting snow albedo, avoiding the model drift into unrealistic states.

In the reference simulation, the mean snow albedo over ice-covered areas varies between 0.65 and 0.8 (Fig. 4a). This represents a substantial reduction compared to simulations assuming fresh and pure snow. In this case the mean albedo is ~0.85, with only tiny variations due to the dependence of snow albedo on the solar zenith angle (Fig. 4a). Most of the time, the reduction of surface albedo by the snow aging effect is larger than the reduction due to dust. Only around the LGM, the dust-induced effect is larger than the pure snow aging effect (Fig. 4a). Geographically explicit surface albedo differences between the different offline experiments are shown in Fig. 5 for a time slice at 15 ka, when ablation reaches its maximum during deglaciation. The reference simulation shows summer albedos as low as 0.4 at the ice sheet margins, where ablation occurs and the snow is old and dust accumulates at the surface while snow is melting (Fig. 5a). Additionally, in localized regions along the margin all snow is melted during summer and bare ice is exposed, which additionally reduces surface albedo, as snow albedo is larger than bare ice albedo over most of the ice sheet because of dust concentrations below ~300 ppmw. Ignoring the effect of snow aging or dust, or both, results in increased surface albedo, mostly along the ice sheet margins (Fig. 5b-d).

The differences in albedo are reflected in the ablation and consequently in the surface mass balance (Fig. 4b,c). When either the snow aging or the dust effect are ignored, the ablation integrated over the NH ice sheets is only ~25 % of the value in the reference simulation (Fig. 4c). This strong reduction in ablation results in a net surface mass balance that is positive throughout the whole last glacial cycle (Fig. 4b).

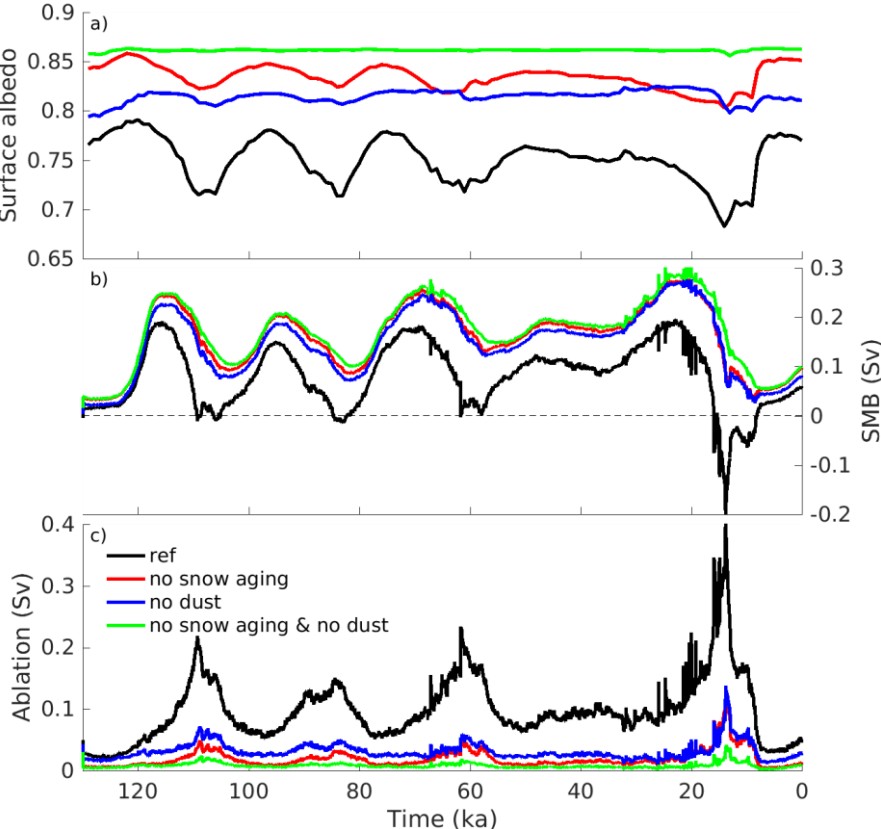

**Figure 4: Results from the offline simulations. a) Mean surface albedo over the area covered by NH ice sheets, b) total surface mass balance and c) total ablation for the experiments indicated in the legend.**

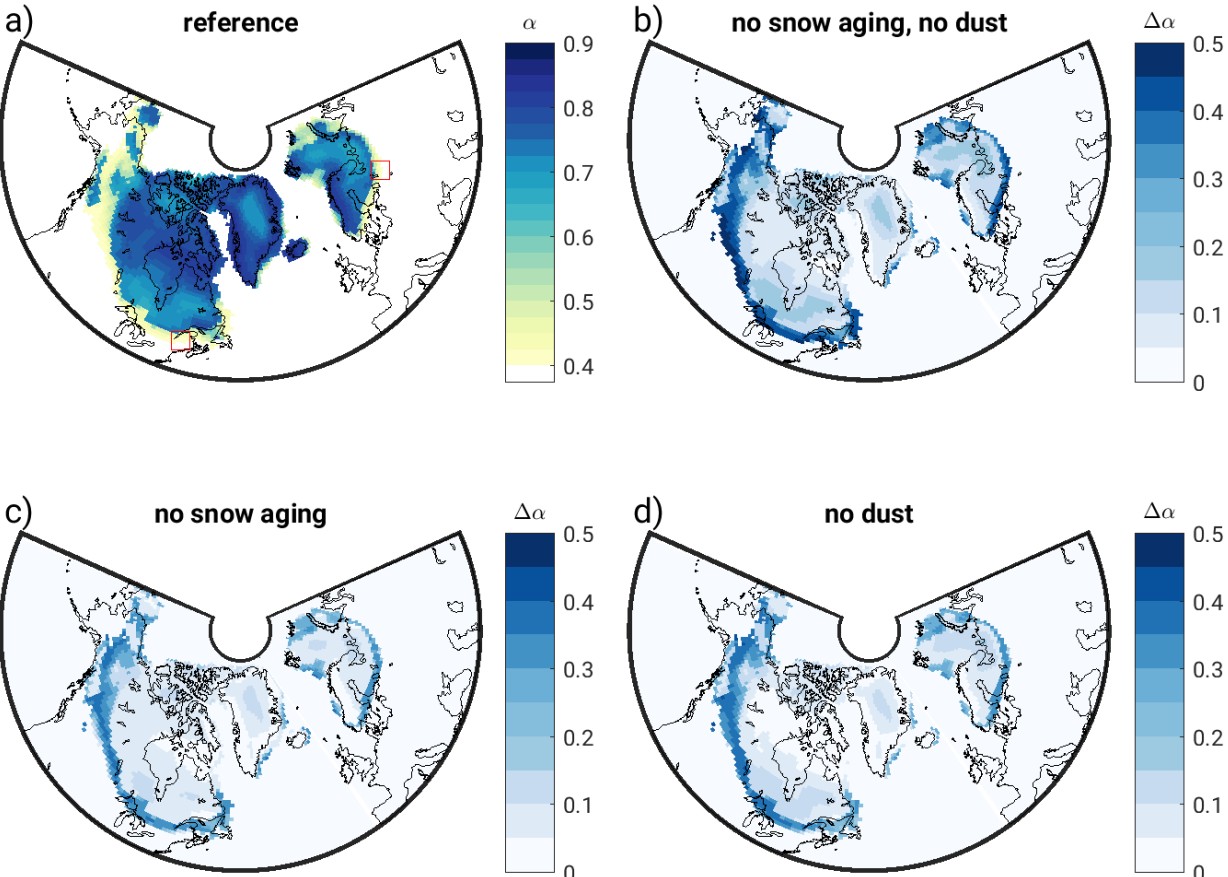

**Figure 5: a) Summer (June-July-August) ice sheets surface albedo at 15 ka for the reference simulation. b-d) Surface albedo anomalies relative to the reference for the three different offline simulations specified in the subplots.**

The albedo of snow is so important for the ice sheet surface energy and mass balance because it strongly controls the length of the snow season and consequently ice melt (Fig. 6). Variations in the albedo of snow during the melt season induced by snow aging and dust content can lead to variations in the length of the snow season of several months (Fig. 6a,b) and consequently to substantial variations in ablation and ice melt (Fig. 6e,f).

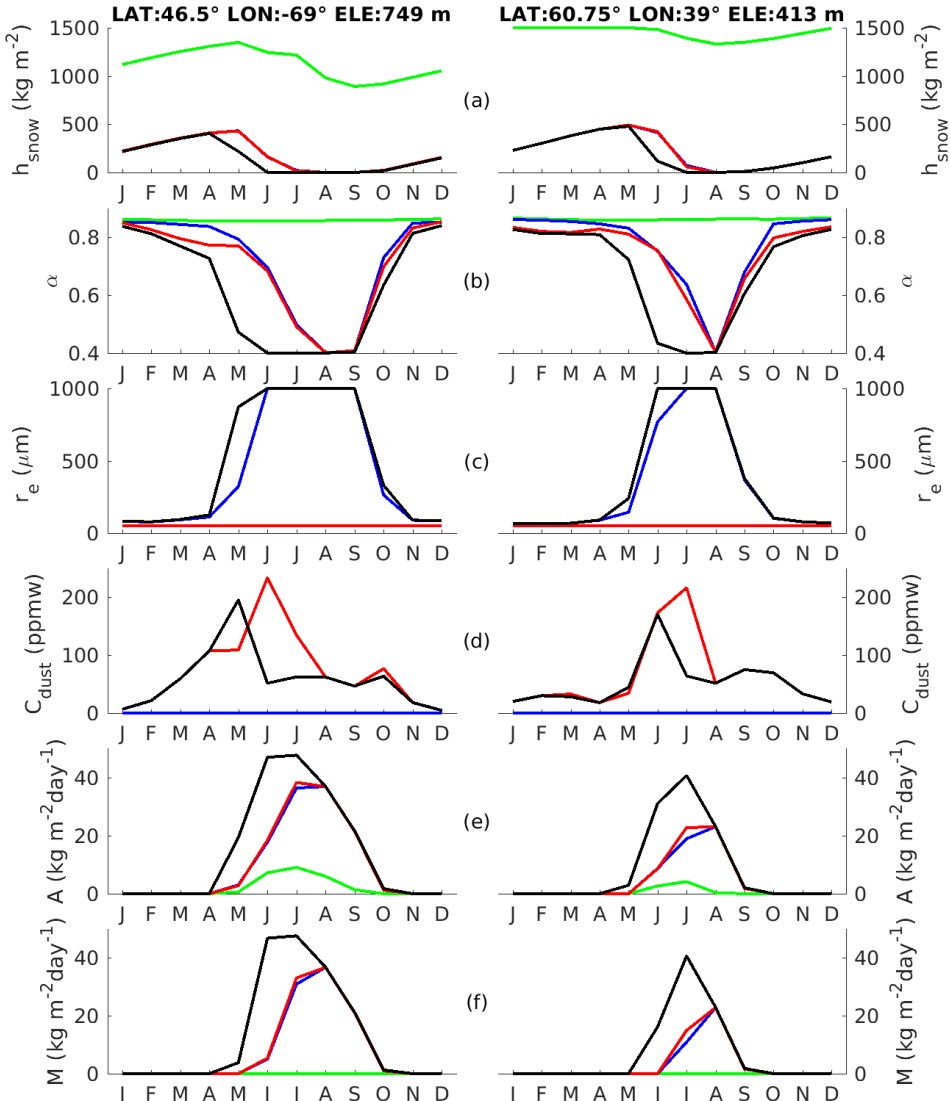

**Figure 6:** Seasonal evolution of several modelled variables at 15 ka at two locations at the southern margin of the Laurentide (left) and Fennoscandian (right) ice sheets for the reference simulation (black lines) and from offline simulations with no dust deposition (blue), no snow aging (red) and no dust deposition and no snow aging (green). The variables shown are a) snow water equivalent, b) surface albedo, c) snow grain size, d) dust concentration in snow, e) ablation and f) ice melt. The location of the two sites is indicated by the red boxes in Fig. 5a.

When the same experiments with and without the effect of snow aging and dust deposition on snow albedo are repeated in the online setup, with the different parameterisations affecting the actual surface energy and mass balance of the ice sheets, the modelled sea level is very different from the reference simulation. In the most extreme case, both the snow aging and the dust darkening effect on snow albedo are ignored. This is equivalent to assuming that snow is always fresh and pure. In this case rapid ice build-up occurs in the model, with sea level dropping below 400 m relative to the present day and with the model subsequently responding only weakly to changes in orbital forcing and greenhouse gases radiative forcing (Fig. 7). Under these conditions, at the LGM ice sheets cover most of Northern America and Eurasia (Fig. 8b). In the experiments where the snow aging or the dust impact on snow albedo are considered separately, excessive ice is grown over North America and a large ice sheet develops over Eurasia (Fig. 8c,d). Also in these experiments, sea level drops well below the estimated LGM value of ~120 m (Fig. 7). Therefore, from the experiments presented and for this model formulation, the effects of dust deposition or snow grain growth acting separately do not allow to simulate a last glacial cycle that is in agreement with climate and sea level reconstructions, because each factor alone is insufficient to prevent glacial inception

over Siberia. Sensitivity experiments that separately ignore the role of dust deposition and aging with a slightly different snow albedo reference value were not performed, but could provide additional insight.

If the snow albedo reduction by deposition of dust produced by glacial sediment transport is ignored, the simulated sea level is very similar to the reference experiment until the LGM. Afterwards, this additional source of dust becomes important to reproduce a full deglaciation in the model (Fig. 7). Dust deposition from glaciogenic origin is negligible compared to aeolian dust over most of the glacial cycle, except during deglaciation, when it becomes comparable or even the dominant source of dust (Fig. 9).

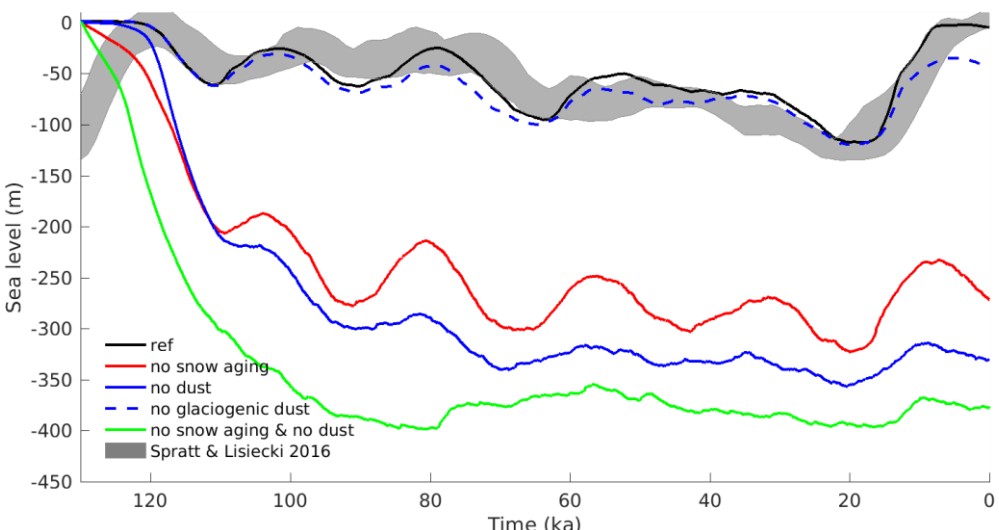

**Figure 7: Effect of snow aging and dust deposition on modelled sea level over the last glacial cycle for different online experiments as indicated in the legend.**

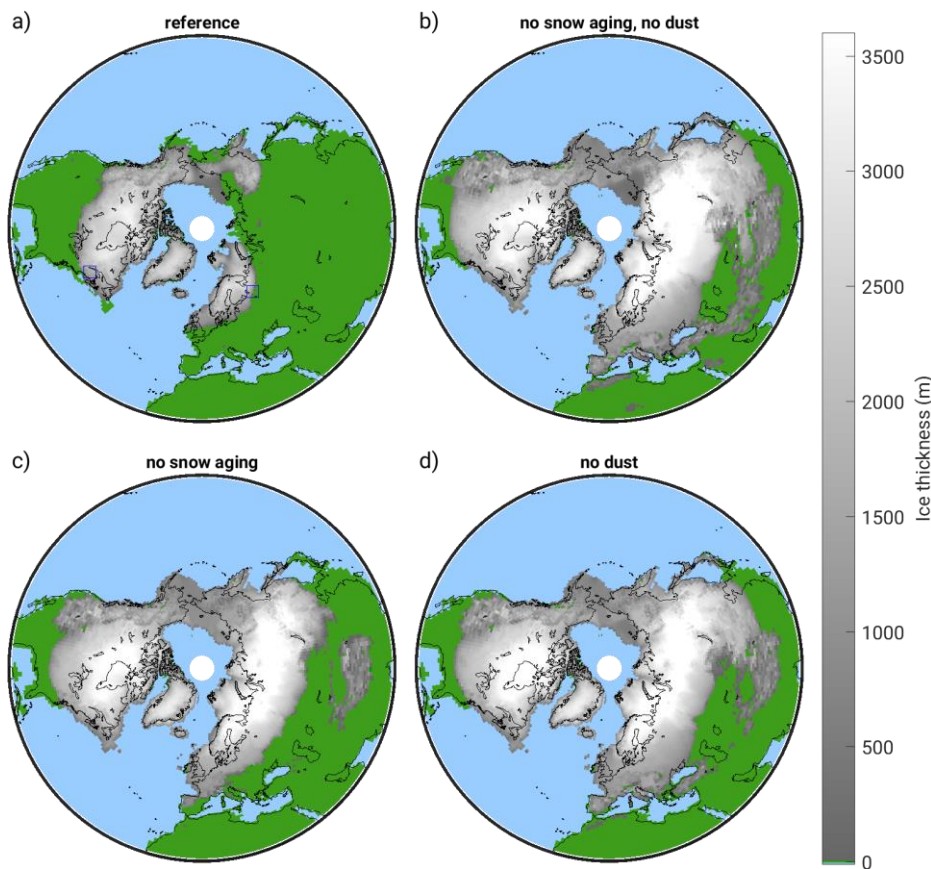

**Figure 8: Ice sheet extent at the last glacial maximum (21 ka) for a) the reference simulation, b) the simulation without snow aging and dust effect on snow albedo, c) the simulation without snow aging and d) the simulation without dust on snow.**

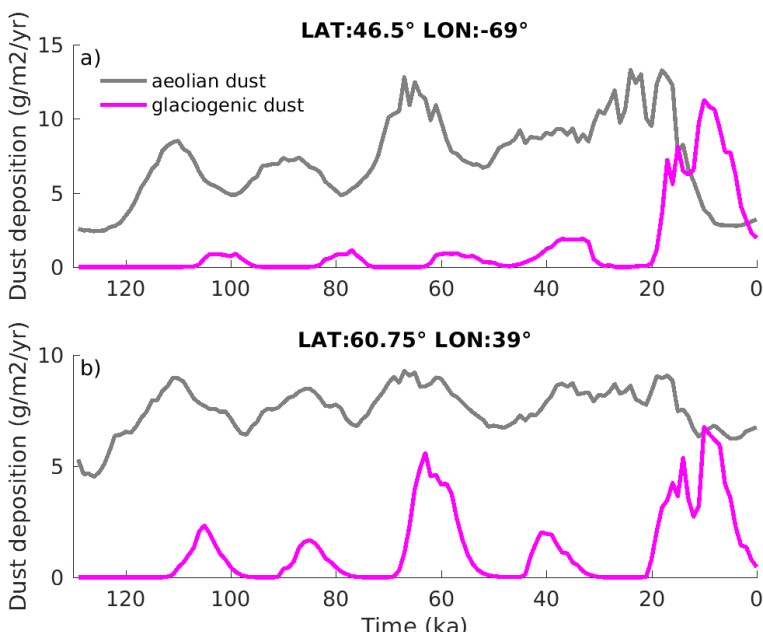

**Figure 9: Evolution of modelled Aeolian (grey) and glaciogenic (magenta) dust deposition at two locations at the southern margin of the LGM Laurentide (top) and Fennoscandian (bottom) ice sheets. The location of the two sites is indicated by the blue boxes in Fig. 8a.**

The choice of the snow albedo scheme also has a considerable impact on the simulated ice volume evolution over the last glacial cycle (Fig. 10a). Using the alternative albedo schemes of Dang et al. (2015) and Gardner and Sharp (2010), which in

general show a weaker darkening effect of dust on snow albedo than the standard scheme used in CLIMBER-2 (Fig. 1), results again in excessive ice growth with LGM ice volume too large by a factor of two (Fig. 10a). However, it is possible that retuning of the model could allow to successfully simulate the last glacial cycle also with these alternative albedo schemes. Conversely, the value used for the albedo of bare ice has a rather limited impact on the simulated glacial cycle (Fig. 10b), because in the model ice ablation is controlled to a large extent by the length of the snow-free season, which is mainly controlled by snow albedo (Fig. 6a). The magnitude of global aeolian dust emissions has a strong control on the modelled glacial cycle evolution. Scaling the dust emissions in the model up or down by a factor of up to 4 leads to a large spread in modelled sea level (Fig. 10c), with simulations with enhanced dust emissions failing to build up enough ice at LGM and simulations with reduced dust emissions leading to excessive ice build up and consequent incomplete deglaciation.

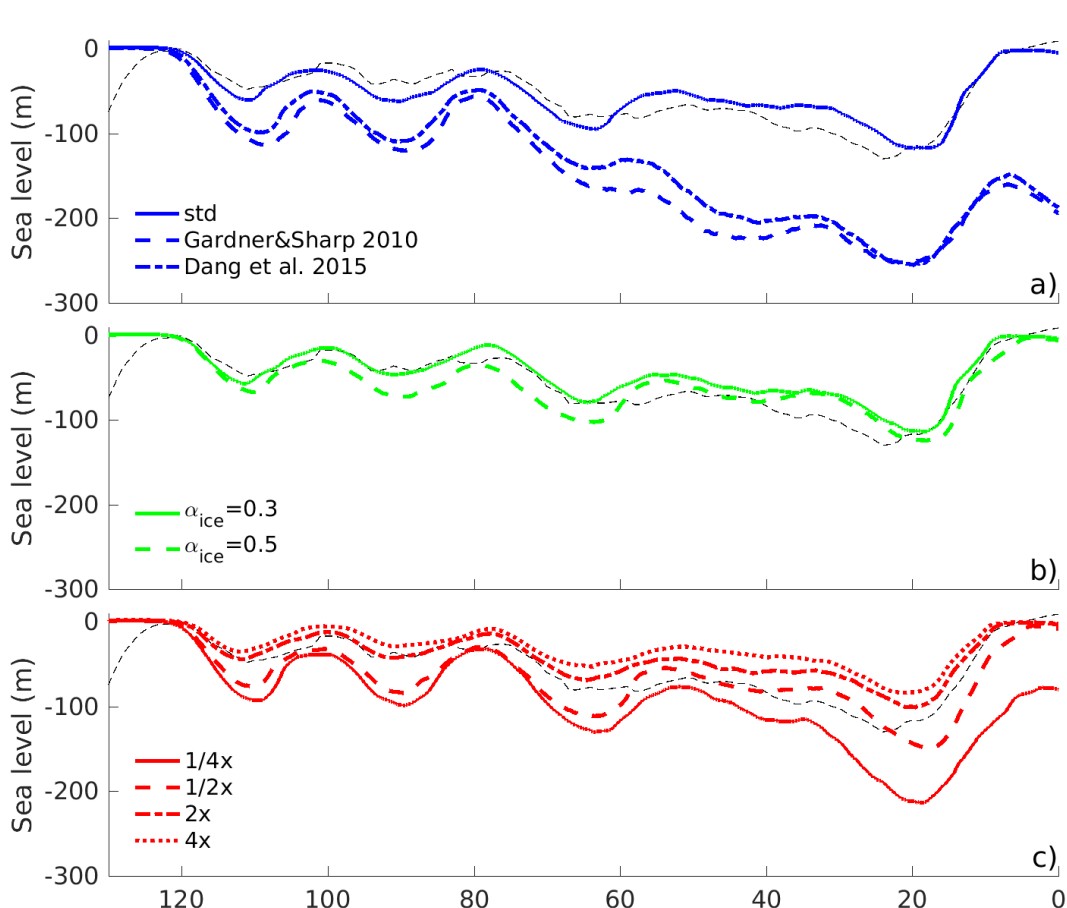

**Figure 10: Uncertainties in modelled sea level evolution over the last glacial cycles resulting from different parameterisations of (a) snow albedo, (b) different values of bare ice albedo and (c) scaling of dust emissions by factors ¼, ½, 2 and 4. The dashed black line represents the sea level reconstruction from Spratt and Lisiecki (2016).**

## 4. Conclusions

In this study we used an Earth system model of intermediate complexity to show that a proper parameterisation of snow albedo over ice sheets is a crucial ingredient for a successful simulation of the last glacial cycle. Both the snow aging effect and the effect of dust deposition on snow albedo play a fundamental role in reducing surface albedo, particularly in the ablation areas. While the snow aging effect on snow albedo is well constrained by observations and theoretical modelling studies, the effect of dust strongly depends on the assumptions about the optical properties of dust. A realistic estimate of the effect of dust on snow albedo does therefore probably have to account for the origin and composition of the dust deposited over ice sheets. Additionally, substantial uncertainties in global and regional dust fluxes during glacial times hinder a quantification of the role of dust darkening of snow for simulating glacial cycles.

**Acknowledgements**

The contribution of Eva Bauer to the offline model setup and her help with the implementation of the fully coupled dust cycle is acknowledged. M.W. acknowledges support by the German Climate Modeling Initiative grant PalMod and by the

German Science Foundation DFG grant GA 1202/2-1.

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
