# Peer review of "The importance of snow albedo for ice sheet evolution over the last glacial cycle"

_Climate of the Past, 2017_

## Referee Comment (RC1) · Anonymous Referee #1 · 1 Dec 2017

Willeit et al. quantify the impact of snow albedo parametrisation (snow aging and dust deposition) for the simulation of the last glacial-interglacial cycle in a model of intermediate complexity. Snow albedo is a crucial parameter for the surface mass balance of the ice sheet and its temporal evolution (over this timescale) is poorly constrained. As such, the study is largely justified. However, in my opinion the manuscript suffers from some important omissions about: study justification/novelty, model description, model validation and methodology discussions (listed below). As it stands the authors mostly show the impact of a higher snow albedo in simulating the last glacial-interglacial cycle.

General comments

- The novelty in this work is not presented in a clear way. The snow albedo is known to be one of a major control on ice sheet surface mass balance and several authors have

already explicitly tested this in coupled ice sheet – climate simulations (e.g. Calov et al. 2005; Bonelli et al. 2009; Helsen et al. 2017; Fyke et al. 2011). Also, it seems difficult to see clearly the difference between this study and the one of Ganopolski et al. (2010), except for the use of an interactive scheme for dust deposition (which is not validated here). Ganopolski et al. (2010) have already shown the importance of dust in their results (Sec. 5.3, quoting): "Hence, at least in our model, accounting for the additional source of dust related to the glacial erosion is crucial for simulating of a complete termination of the glacial cycle [...]". Given that: i) the conclusions in Willeit et al. (2017) are almost identical to the one of Ganopolski et al. (2010) and; ii) there is no real improvement in the model; I feel like the study needs a stronger justification.

- Methodology. The model has been tuned to reproduce the glacial-interglacial cycles for a specific snow parametrisation. I am thus not surprised that the omission of one process affecting the snow albedo lead to an erroneous ice sheet evolution. Are the authors testing the actual processes (dust/aging) or simply the value of the albedo? Switching between off and on the two processes with the same value for the fresh snow albedo is an unjustified oversimplification. They could have tried to retune the model without the aging and/or dust (considering a perpetual "dirty fresh snow" for example): if they were able to show that it is impossible to get a realistic ice sheet volume evolution in doing so, then they might have claimed that aging/dust are clearly important. In addition, if the real novelty of this work is to use the interactive dust, they should have shown the difference of their model compared to the prescribed dust version of Ganopolski et al. (2010).

- Validation of the scheme: the study could be more convincing if the aging and dust parametrisations were validated against observations or state-of-the-art model simulations. In particular, it could be useful to see if the scheme reproduces the seasonal variations of albedo of the Greenland ice sheet and high latitudes regions. Again, as it stands, the reader is left with the impression that the parametrisations have been chosen (tuned) to reproduce the last glacial-interglacial cycle. As a result it is obvious

that the model will not work if the processes are not included. I would also like to see how well the dust deposition changes over Greenland along the cycle simulated by the model compares to actual dust in Greenland ice cores.

- Model presentation: the albedo computation in the model should be presented in this paper, in particular on how age and dust relate to albedo. The reader has to check the appendix of Calov et al. 2005 to get more information on this. Also, the same is true for the dust from glaciogenic sediments, for which the authors only state that the model accounts for. I also think that more information on the SEMI model could be added (which variables are bilinearly interpolated? Which variables depend on sub-grid topography? Etc.).

Specific comments

P2L1 In fact, algae could be more important than non-algal impurities for bare ice albedo (Musilova et al. 2016; Stibal et al. 2017).

P2L23-24 SEMI does not perform a physically based downscaling of climatological fields. Most of the variables are bilinearly interpolated. The SMB is physically based though.

P2L27-28 Does this include the Antarctic ice sheet?

P3L5 How this is computed? Can you really differentiate between the two types of dust? Please expand on this.

P3L7 On which grid are you looking that? The SICOPOLIS grid or the native atmospheric grid?

P3 Eq.X? Is this calculated on each atmospheric timestep? What is the value of the atmospheric timestep?

P4 Fig 1 The CLIMBER2 albedo presented here is the one tuned to reproduce the glacial interglacial cycles and they are systematically below the values of Dang et al.

and Gardner and Sharp ( 0.1 difference, except for pure snow where it is very close). This is not surprising that using one of Dang et al. or Gardner and Sharp results in an overestimation of the ice volume.

P4L12 Antarctic kept constant to present-day observations?

P5 Figure 2 Why the surface mass balance is increasing over the Holocene whilst the ice volume remains constant?

P6L19-20 You state in line 8 that the simulated dust deposition is roughly 3000 Tg/yr but later you mention the scaling factor to get 3000 Tg/yr. This is confusing: what is the actual simulated dust deposition before the application of the scaling factor? Using an other scaling factor you could end up with significant study conclusions (maybe for more dust, you might no need the snow aging to reproduce the cycles. . .).

P6 Fig 3 This is only aeolian dust? Do Mahowald and Lambert include the glaciogenic dust as well?

P6 Fig 3 It could be nice to have the extent of the (observed) ice sheets for the two time periods on this plot.

P7L8 The dust effect is larger at the LGM because you have the contribution from the glaciogenic sediments?

P7L13 I do not understand this statement: your ice albedo is 0.4 and your old/dirty snow has an albedo which can be lower than 0.4.

P8 Fig 5 Why there is a corridor of low albedo values between the coastal grid points of the gulf of Alaska and the rest of the Laurentide ice sheet?

P8 Fig 5 Is the depicted albedo only for grid points covered by an ice sheet? If not, please add the extent of the simulated ice sheet in this.

P9 Fig 6 Does the surface mass balance scheme includes any kind of refreezing?

P9 L17-20 [...] "From the experiments presented and for this model formulation" should be added. Again, I am not convinced that you actually test the actual processes. Extrapolating: the use of Dang et al. and Gardner and Sharp does not allow for a realistic cycle neither, does this mean that there is still a missing process like algae?

P10 L1 How this is computed? Can we really distinguish this from the rest of the dust? It could be useful to have a map of this.

P10 L3-4 Is it fair to say that this is a tunable additional source of dust in order to produce a realistic cycle?

P10 L3-4 Your maximum dust over the whole cycle is at about 15k thanks to this glaciogenic dust. It seems important to clearly state where does this come from and why this process only appears at the end of the cycle. A few maps at selected snapshots could be nice for albedo, dust and SMB.

P11 L10-12 "ice is covered by snow most of the year, even in net ablation areas" To melt the ice sheet you need to melt the ice in summer, using the ice albedo. Can you give more explanation on why the ice albedo is not playing in your deglaciation scenario?

P11 L12-16 These experiments are interesting, to my opinion. It would have been nice to see these experiments combined with the omission of snow aging. Increased dust but no aging might produce a realistic cycle? Or combining dust deposition scaling factor with Dang et al. or Gardner and Sharp parametrisation.

General

- What about the sea ice albedo? Do you have a similar scheme that includes dust and aging? If not, why.

- The computed SMB is a function of albedo, but also it depends on the other energy balance terms. In particular, how transparent your clouds are is probably very important for the surface mass balance. Snow albedo is always crucial for the SMB, but you might

not need dust to reproduce the cycle for different parametrisation of clouds.

Technical corrections

P9 Fig 6 The location of the two sites are not indicated by black boxes.

P9 L15 Replace Nord by Northern

P9 L16 && L18 Separately instead of "in isolation"

P11 Fig 9 there is no blue boxes in Fig 5.

Bibliography

Bonelli, S., Charbit, S., Kageyama, M., Woillez, M.-N., Ramstein, G., Dumas, C., and Quiquet, A.: Investigating the evolution of major Northern Hemisphere ice sheets during the last glacial-interglacial cycle, Clim. Past, 5, 329–345, doi:10.5194/cp-5-329-2009, 2009.

Fyke, J. G., Weaver, A. J., Pollard, D., Eby, M., Carter, L., and Mackintosh, A.: A new coupled ice sheet/climate model: description and sensitivity to model physics under Eemian, Last Glacial Maximum, late Holocene and modern climate conditions, Geosci. Model Dev., 4, 117-136, https://doi.org/10.5194/gmd-4-117-2011, 2011.

Helsen, M. M., van de Wal, R. S. W., Reerink, T. J., Bintanja, R., Madsen, M. S., Yang, S., Li, Q., and Zhang, Q.: On the importance of the albedo parameterization for the mass balance of the Greenland ice sheet in EC-Earth, The Cryosphere, 11, 1949-1965, https://doi.org/10.5194/tc-11-1949-2017, 2017.

Musilova M, Tranter M, Bamber JL, Takeuchi N, Anesio AM. Experimental evidence that microbial activity lowers the albedo of glaciers. Geochem Perspect Lett., doi:10.7185/geochemlet.1611, 2016.

Stibal M, Box JE, Cameron KA, Langen PL, Yallop ML, Mottram RH, et al. Algae drive enhanced darkening of bare ice on the Greenland ice sheet. Geophysical Research

Letters, 44, doi:10.1002/2017GL075958, 2017.

---

## Referee Comment (RC2) · J. Alvarez-Solas (Referee) · 8 Jan 2018

Willeit and Ganopolski show the importance of considering the effects of snow aging and dust on the snow albedo and consequently on satisfactorily simulating glacial cycles. The article is well written and its relevance is properly justified. In my opininon, the novely of the paper does not lie directly on the results but on the presentation of the parameterizations for accounting on the mentioned effects on snow albedo. Accordingly, the main weakness of the study is reproducibility. The authors should expand on the snow albedo parameterization in order to other groups being able to reproduce (and benefit from) the current study.

General comments

[Figure]

About reproducibility:

Ice sheet – climate coupling represents a considerable ongoing effort for modeling groups. The authors of this article have already convincingly shown in previous studies the necessity of accounting for the snow albedo reduction from ice aging and dust in order to succesfully simulate a deglaciation. This article furtherly contributes to this idea and presents the needed albedo parameterizations to do so. This later aspect can be of great importance to groups currently starting to couple GCMs to thermodynamical ice sheet models. Thus, these parameterizations need to be accordingly described.

1. In page 3, line 14, the snow age factor parameterization is described:

1.1 It might be obvious, but the reader could wonder whether the aging of the snow can simply be computed as a function of temperature and snowfall. Please, ellaborate on this and add references.

1.2 The definition of $T_0$ is missing.

1.3 The age factor is used to represent the grain size. And Fig.1 shows grain radius. How is CLIMBER-2 translating each other? It is linear? Please provide the related expression.

1.4 Fig.1: Besides the pure snow case, CLIMBER-2 seems to be underestimating the albedo compared to the two other parameterizations. Why? A potential explanation is given by the sentence: "... explained by the choice of the imaginary refractive index of dust". Please, be more specific. On the other hand, the effect of the alternative parameterizations on simulating the glacial cycle is described in the Results section, but it s not explained. I imagine this can simply be a matter of "tunning". Re-calibrating the age factor (or other components of the model) for the two alternative approaches will produce a succesfull ice-volume evolution. If this is the case, please aknowledge in the paper. Otherwise, the reader remains wondering about the realism of the different approaches.

2. In page 10, the effects of considering aeolian and glaciogenic dust individually are discussed. The interactive aeolian dust representation is conveniently described in previous studies. I could not, however, find the equivalent for glaciogenic dust. How is glaciogenic dust generated in CLIMBER-2? Please provide the necessary information. Furthermore, when Fig.7 shows glaciogenic dust as a necessary condition for a full deglaciation.

About discussing the necessity of including a dust cycle:

In the Conclusions section it can be read: "In this study we used an Earth system model of intermediate complexity to show that a proper parameterisation of snow albedo over ice sheets is a crucial ingredient for a successful simulation of the last glacial cycle." This and previous studies from these authors support this conclusion. Nevertheless, other models/groups have shown succesfull glacial cycle simulations without the necessity to invoke "a proper parameterisation of snow albedo". For example, in Abe-Ouchi et al 2007 CP and 2013 Nature, the ablation-isostatic adjustment feedback together with elevation and other feedbacks appear to represent enough processes to simulate the deglaciation.

The current main conclusion (see above) of this paper give rise to interesting related questions: Could CLIMBER-2 simulate a deglaciation without considering the effects of dust on snow albedo? If affirmative, which are then the key processes? Are those other processes equally realistic? Is all the relevant physichs necessary for understanding deglaciations already contained in EMICs? ... I understand that the authors could see these questions as out of the scope for the current article, but I also believe the readers will appreciate further the current paper if a discussion on this aspect is included.

Specif comments:

Page 1, line 10 and 14: Please use "light-absorbing ..." as later in the paper.

Page 3, line 8: add "in" afer "snow albedo used..."

Caption figure 9: erratum: glaciogenic

---

## Author Comment (AC1) · 12 Feb 2018

**Response to Anonymous Referee #1**

Willeit et al. quantify the impact of snow albedo parametrisation (snow aging and dust deposition) for the simulation of the last glacial-interglacial cycle in a model of intermediate complexity. Snow albedo is a crucial parameter for the surface mass balance of the ice sheet and its temporal evolution (over this timescale) is poorly constrained. As such, the study is largely justified. However, in my opinion the manuscript suffers from some important omissions about: study justification/novelty, model description, model validation and methodology discussions (listed below). As it stands the authors mostly show the impact of a higher snow albedo in simulating the last glacial-interglacial cycle.

We would like to thank the reviewer for his constructive comments on our manuscript. We have responded to the issues raised by the reviewer below. The original reviewer comments are in black, our responses in blue.

**General comments**

- The novelty in this work is not presented in a clear way. The snow albedo is known to be one of a major control on ice sheet surface mass balance and several authors have already explicitly tested this in coupled ice sheet – climate simulations (e.g. Calov et al. 2005; Bonelli et al. 2009; Helsen et al. 2017; Fyke et al. 2011). Also, it seems difficult to see clearly the difference between this study and the one of Ganopolski et al. (2010), except for the use of an interactive scheme for dust deposition (which is not validated here). Ganopolski et al. (2010) have already shown the importance of dust in their results (Sec. 5.3, quoting): "Hence, at least in our model, accounting for the additional source of dust related to the glacial erosion is crucial for simulating of a complete termination of the glacial cycle [...]". Given that: i) the conclusions in Willeit et al. (2017) are almost identical to the one of Ganopolski et al. (2010) and; ii) there is no real improvement in the model; I feel like the study needs a stronger justification.

The reviewer is absolutely right – the importance of snow and ice albedo parameterizations for ice sheets modeling has been known for long time. To the contrary, the role of snow albedo parameterization on modeling of glacial cycles is much less understood and so far was only shortly discussed in several of our papers. Most of other simulations of glacial cycle(s), like the above cited Bonelli et al. (2009) as well as Tarasov and Richard Peltier (2002), Zweck and Huybrechts (2005), Charbit et al. (2007), Abe-Ouchi et al. (2007), Lunt et al. (2008), Gregoire et al. (2012), Liakka et al. (2016) and many others employed the so-called Positive Degree Day scheme which does not even account explicitly for snow and ice albedos. Indeed, in Ganopolski et al. (2010) we concluded that the dust (of glaciogenic origin) plays an important role in termination of glacial cycle. But this paper contains only one figure (Fig. 11c) demonstrating the effect of dust deposition on simulated glacial cycle, and only for the dust originated from glacial erosion. In Ganopolski et al (2010) we did not perform a systematic analysis of the effect of different parameterizations of the dust darkening effect on snow albedo, how important snow aging is and how sensitive the model is to the total amount of dust which is deposited on the ice sheets. Why such analysis became so important now? Because of the major development in paleoclimate modeling, namely because it became possible to use complex Earth system Models based on GCMs to test Milankovich theory and simulate the last glacial cycle. One of such ambitious projects is the German National Modeling Initiative PalMod (Latif

et al. 2016) from which this work is partly funded. In the framework of this and similar projects over-simplistic obsolete scheme like PDD will be substituted by a physically based energy balance approach similar to that used in CLIMBER-2. However, unlike CLIMBER-2, complex ESMs are extremely computationally expensive and cannot afford to use the try and error method. In fact, they can only perform a single simulation of one glacial cycle. This implies that the crucial model parameters and parameterizations have to be properly calibrated before launching long-term simulations. Our study clearly shows that the proper parameterizations of snow albedo which includes snow aging effect, effect of impurities and the synergy between both, is absolutely crucial for successful simulation of glacial cycle. Even the choice of different parameterizations for the effect of dust on snow albedo can lead to very different outcomes of glacial cycle simulations. We believe, these findings are important and will significantly facilitate simulations of glacial cycles with complex ESMs.

Concerning "real improvement in the model". The aim of introducing of the fully interactive dust cycle in CLIMBER-2 is not to improve model results if under "improvement" the reviewer understands better agreement between model and data. This agreement is already good in Ganopolski et al. (2010). Development of a comprehensive Earth system model where all important processes are included and all components (climate, ice sheet, carbon and dust cycles) are fully interactive is necessary for the decisive test of Milankovitch theory by simulating realistic glacial cycles by using orbital forcing as the only prescribed forcing.

- Methodology. The model has been tuned to reproduce the glacial-interglacial cycles for a specific snow parametrisation. I am thus not surprised that the omission of one process affecting the snow albedo lead to an erroneous ice sheet evolution. Are the authors testing the actual processes (dust/aging) or simply the value of the albedo? Switching between off and on the two processes with the same value for the fresh snow albedo is an unjustified oversimplification. They could have tried to retune the model without the aging and/or dust (considering a perpetual "dirty fresh snow" for example): if they were able to show that it is impossible to get a realistic ice sheet volume evolution in doing so, then they might have claimed that aging/dust are clearly important. In addition, if the real novelty of this work is to use the interactive dust, they should have shown the difference of their model compared to the prescribed dust version of Ganopolski et al. (2010).

The fact that CLIMBER-2 is capable of reproducing glacial-interglacial cycles is the result of many years of work, which included model improvements, inclusion of new processes based on new process understanding and obviously also tuning of unconstrained model parameters. Even with a relatively fast model like CLIMBER-2 it is practically impossible to explore the whole parameter space and it could well be that other parameter combinations could lead to a successful simulation of glacial cycles. This is beyond the scope of the present study. The goal of this study is to clearly show that surface albedo plays a crucial role for ice sheet evolution and not only do fundamental processes affecting snow albedo, such as snow aging and dust deposition, play an important role, but even using slightly different parameterisations of the same process (e.g. the effect of mineral dust impurities on snow albedo) can lead to qualitatively very different results. This is a clear message to readers who are interested in simulating glacial cycles and more in general the long-term evolution of ice sheets.

Concerning differences between new results and Ganopolski et al. (2010). They exist of course, but they are not significant. As has been explained above, the aim for substitution of prescribed spatial

patterns of dust deposition based on GCM time slice simulations used in our earlier studies, by the dust deposition fields simulated by our own dust cycle component is not to "improve" model performance but to design a fully interactive and internally consistent Earth System model suitable for testing Milankovitch theory.

- Validation of the scheme: the study could be more convincing if the aging and dust parametrisations were validated against observations or state-of-the-art model simulations. In particular, it could be useful to see if the scheme reproduces the seasonal variations of albedo of the Greenland ice sheet and high latitudes regions. Again, as it stands, the reader is left with the impression that the parametrisations have been chosen (tuned) to reproduce the last glacial-interglacial cycle. As a result it is obvious that the model will not work if the processes are not included. I would also like to see how well the dust deposition changes over Greenland along the cycle simulated by the model compares to actual dust in Greenland ice cores.

The model parameterization has been developed based on existing schemes (BATS (Dickinson et al., 1986)) which in turn were calibrated against observational data. Direct comparison of CLIMBER-2 results with observational data is not very useful because of coarse resolution and schematic geography. However the newest land scheme for the next CLIMBER model (Willeit and Ganopolski, 2016), which employs a similar albedo parameterization but has a much higher spatial resolution shows a good agreement of simulated albedo with empirical data (Fig. 5 in Willeit and Ganopolski (2016)). Concerning the comparison of seasonal variations of Greenland albedo with observations, such a comparison does not make much sense because the elevation, temperature, precipitation, surface radiation, etc. in model simulations differ from reality and all these factors (not just the parameterization of albedo) affect surface albedo. Comparison can be made in specially designed off-line simulation where elevation and climatology are prescribed from observation and only albedo is computed. We will present such an experiment in the revised version of the paper and compare it with observations.

 I would also like to see how well the dust deposition changes over Greenland along the cycle simulated by the model compares to actual dust in Greenland ice cores.

We discussed the dust model performance against data and other models in Bauer and Ganopolski (2010, 2014). This comparison shows that the model is doing reasonably well when data uncertainty and big discrepancies between different state of the art models are taken into consideration. We did not show model performance for Greenland for several reasons. Firstly, even during glacial time observed dust deposition over Greenland was very low (order of 0.1 g/m2/a) and had no impact on Greenland ice sheet mass balance. Second, the main dust sources for Greenland are located in Central Asia, which implies an extremely long atmospheric dust transport along very complex orography.  Obviously, the coarse-resolution CLIMBER-2 is not good for modeling such process. Third, the model grid cell which incorporates the real Greenland is only by 50% covered by ice sheet and by 50% by ocean. Therefore the averaged elevation of this model grid cell is very different from the elevation of Greenland summit. Similarly, the average precipitation rate over the entire "Greenland grid cell" is an order of magnitude higher than precipitation at the locations of the ice cores. All this makes comparison between modeled and ice core data not very useful.

- Model presentation: the albedo computation in the model should be presented in this paper, in particular on how age and dust relate to albedo. The reader has to check the appendix of Calov et al. 2005 to get more information on this. Also, the same is true for the dust from glaciogenic sediments,

for which the authors only state that the model accounts for. I also think that more information on the SEMI model could be added (which variables are bilinearly interpolated? Which variables depend on subgrid topography? Etc.).

Although all aspects of albedo parameterization, dust and surface mass balance simulations have been given in our previous papers (Bauer and Ganopolski, 2010; Calov et al., 2005; Ganopolski et al., 2010, etc…), following reviewer's suggestion we will added a more complete description of the snow albedo parameterization, dust cycle and SEMI model.

**Specific comments**

P2L1 In fact, algae could be more important than non-algal impurities for bare ice albedo (Musilova et al. 2016; Stibal et al. 2017).

We have included references to these two papers in the revised manuscript.

P2L23-24 SEMI does not perform a physically based downscaling of climatological fields. Most of the variables are bilinearly interpolated. The SMB is physically based though.

This is not correct. Importantly, precipitation is downscaled accounting for the slope effect and the desert-elevation effect. Radiation and atmospheric temperature and humidity are first interpolated bilinearly and then corrected for the surface elevation of the ice sheet.

P2L27-28 Does this include the Antarctic ice sheet?

No, Sicopolis is applied only to the Northern Hemisphere.

P3L5 How this is computed? Can you really differentiate between the two types of dust? Please expand on this.

To clarify this point we added the following paragraph in the revised paper:
*"This dust source is not included in the global dust cycle model due to its very local origin, which can not be represented on the coarse atmospheric grid. Dust deposition produced from glaciogenic sources is parameterized based on the assumption that the emission of glaciogenic dust is proportional to the delivery of glacial sediments to the edge of an ice sheet (see Ganopolski et al. 2010 Appendix A for details). Most of the glaciogenic dust originates from the southern flanks of the ice sheets and this source is significant only for mature ice sheets, which reach well into areas covered by thick terrestrial sediments."*

P3L7 On which grid are you looking that? The SICOPOLIS grid or the native atmospheric grid?

The snow albedo parameterisations are applied both to the surface scheme on the atmospheric grid and to SEMI on the ice sheet grid.

P3 Eq.X? Is this calculated on each atmospheric timestep? What is the value of the atmospheric timestep?

The snow age factor is computed on each atmospheric timestep and the timestep is one day.

P4 Fig 1 The CLIMBER2 albedo presented here is the one tuned to reproduce the glacial interglacial cycles and they are systematically below the values of Dang et al. and Gardner and Sharp ( 0.1

difference, except for pure snow where it is very close). This is not surprising that using one of Dang et al. or Gardner and Sharp results in an overestimation of the ice volume.

We agree that it is not surprising that using one of Dang et al. or Gardner and Sharp results in a larger simulated ice volume. However, it is not obvious a priori that using these alternative albedo schemes results in a total failure to simulate the last glacial cycle. The snow albedo for the different schemes shown in Fig.1 is substantially different only when dust is involved. Hence, as dust deposition varies strongly in space and time, the albedo difference between the different schemes will also be a complex function of space and time.

P4L12 Antarctic kept constant to present-day observations?

Sicopolis is applied only to the NH and Antarctica is kept constant to present day observations. This will be explicitly stated in the revised version of the manuscript.

P5 Figure 2 Why the surface mass balance is increasing over the Holocene whilst the ice volume remains constant?

The increasing SMB is related to the decrease of Greenland melt during the Holocene and is almost completely balanced by increased ice calving to the ocean and therefore not seen in global ice volume.

P6L19-20 You state in line 8 that the simulated dust deposition is roughly 3000 Tg/yr but later you mention the scaling factor to get 3000 Tg/yr. This is confusing: what is the actual simulated dust deposition before the application of the scaling factor? Using an other scaling factor you could end up with significant study conclusions (maybe for more dust, you might no need the snow aging to reproduce the cycles...).

In the revised paper we will make clear on line 8 that the 3000 Tg/yr deposition is a result of adjusting the dimensionless global calibration constant cq in Eq. (8) of Bauer and Ganopolski (2010). The cq factor is part of the original dust model (cq = 0.0212) and has been slightly increased in the present study (cq = 0.03) to get present global dust emissions of 3000 Tg/yr. The effect of scaling dust emissions up or down from the reference value on simulated glacial cycles is illustrated in Fig. 10.

P6 Fig 3 This is only aeolian dust? Do Mahowald and Lambert include the glaciogenic dust as well?

The modelled dust deposition includes both aeolian and glaciogenic dust. The same is true for the Lambert dataset, while Mahowald 1999 does not include glaciogenic dust (Mahowald 2006 does, but is not shown here). This will be specified in the caption of Fig. 3 in the revised manuscript.

P6 Fig 3 It could be nice to have the extent of the (observed) ice sheets for the two time periods on this plot.

We will add the modeled (observed) ice sheet extent to the figure.

P7L8 The dust effect is larger at the LGM because you have the contribution from the glaciogenic sediments?

No, the glaciogenic dust starts to become important after LGM, during deglaciation (see e.g. Fig. 9). The larger effect of dust on albedo at LGM is a result of larger aeolian dust emissions at LGM compared to other times (Fig. 2g) combined with ice sheet extent reaching further south where dust deposition is larger.

P7L13 I do not understand this statement: your ice albedo is 0.4 and your old/dirty snow has an albedo which can be lower than 0.4.

This is correct, but most of the time the dust concentration in the snow is fairly low (below ~300 ppmw) and therefore the albedo of snow is usually larger than the albedo of ice. This has been added to the paper.

P8 Fig 5 Why there is a corridor of low albedo values between the coastal grid points of the gulf of Alaska and the rest of the Laurentide ice sheet?

Because of lower ice elevation along the corridor.

P8 Fig 5 Is the depicted albedo only for grid points covered by an ice sheet? If not, please add the extent of the simulated ice sheet in this.

As mentioned in the figure caption the albedo is only shown for ice covered grid points.

P9 Fig 6 Does the surface mass balance scheme includes any kind of refreezing?

No, refreezing is not accounted for in the SMB scheme.

P9 L17-20 [...] "From the experiments presented and for this model formulation" should be added. Again, I am not convinced that you actually test the actual processes. Extrapolating: the use of Dang et al. and Gardner and Sharp does not allow for a realistic cycle neither, does this mean that there is still a missing process like algae?

The sentence has been modified as suggested.
There is no obvious reason to assume that the albedo parameterisations of Dang et al. and Gardner and Sharp are better or more realistic than the original CLIMBER-2 scheme based on Warren and Wiscombe (1980). They basically differ in the strength of the dust effect on snow albedo, which mainly depends on the assumed dust radiation absorption properties. A number of different papers have recently shown that algae could play an important role and in the future we will consider the possibility to represent this effect in the model.

P10 L1 How this is computed? Can we really distinguish this from the rest of the dust? It could be useful to have a map of this.

This is just the glaciogenic dust that is described in the paper, with some more details in the revised version as explained in the response to a reviewers question above. Figure 9 gives already an idea of the temporal evolution of glaciogenic versus aeolian dust and we think that a map would not add much more information here. A spatially explicit representation of the glaciogenic dust production can be found in Ganopolski et al. (2010), Fig. 9c.

P10 L3-4 Is it fair to say that this is a tunable additional source of dust in order to produce a realistic cycle?

No, it is not. This (glaciogenic) source of dust is not "tunable"- it is real. The DIRTMAP data (Kohfeld and Harrison, 2001) show that dust deposition near to the southern margins of the Laurentide and Fenoscandian ice sheets during LGM are typically within the range 50-200 g/m2/a, which is one or even two orders of magnitude large than the model can simulate without taking into consideration glaciogenic dust sources (i.e. Mahowald et al. (2006)). Note that the dust deposition rates at the LGM in the same areas in CLIMBER simulations (Fig. 10 in Ganopolski et al. (2010)) are about 50 g/m2/s. Therefore there is no reason to believe that the effect of glaciogenic dust is overestimated in our simulations.

P10 L3-4 Your maximum dust over the whole cycle is at about 15k thanks to this glaciogenic dust. It seems important to clearly state where does this come from and why this process only appears at the end of the cycle. A few maps at selected snapshots could be nice for albedo, dust and SMB.

More details on the origin of the glaciogenic dust have been included in the revised version of the paper. This, together with Fig. 7 and Fig. 9 should be sufficient to allow the reader to better understand where it comes from and where and when it is important.

P11 L10-12 "ice is covered by snow most of the year, even in net ablation areas" To melt the ice sheet you need to melt the ice in summer, using the ice albedo. Can you give more explanation on why the ice albedo is not playing in your deglaciation scenario?

Yes, but the length of the (snow-free) ice melt season is controlled by snow albedo and in the model the length of the melt season is more important than the actual value of ice albedo.

P11 L12-16 These experiments are interesting, to my opinion. It would have been nice to see these experiments combined with the omission of snow aging. Increased dust but no aging might produce a realistic cycle? Or combining dust deposition scaling factor with Dang et al. or Gardner and Sharp parametrisation.

Indeed, the right results can be obtained for the wrong reason. We cannot rule out a possibility that combination of a model version without snow aging effect with excessive dust deposition can lead to reasonably realistic results. But what one can learn from that if we know that the snow aging effect on albedo is real and important? As we explained above, the purpose of this study is to contribute to successful simulation of glacial cycle with complex Earth system models. These models do account for aging effect on albedo but not yet account for the effect of dust and the synergy between the two. Our study shows that all three (aging, dust and synergy) are important.

**General**

- What about the sea ice albedo? Do you have a similar scheme that includes dust and aging? If not, why.

Modeling albedo of sea ice even more complex than albedo of snow and it can be done properly with a high spatial resolution model. Note that the resolution of climate component of the CLIMBER-2 model (unlike the resolution of ice sheet model and SEMI) is very coarse – 10°x51°. This is why we use a simple parameterization for sea ice albedo, where the later depends only on surface temperature. The reason why we ignore the effect of dust deposition on sea ice albedo is obvious: the dust deposition over areas covered by sea ice even during the LGM was very low,  typically

1g/cm2/a or less, which is two orders of magnitude smaller that the dust deposition rate over the southern flanks of Laurentide and Eurasian ice sheets.

- The computed SMB is a function of albedo, but also it depends on the other energy balance terms. In particular, how transparent your clouds are is probably very important for the surface mass balance. Snow albedo is always crucial for the SMB, but you might not need dust to reproduce the cycle for different parametrisation of clouds.

According to our simulations, dust deposition is important. We would not be surprised if future studies would show that the dust deposition is less (or more) important for glacial cycles than we report here. After all, even the equilibrium climate sensitivity is known only with the accuracy of 50%. However, it is very unlikely that clouds alone will make the job. In spite of relatively simple parameterizations, CLIMBER-2 simulates planetary albedo and atmospheric energy balance in a good agreement with observational data (Petoukhov et al., 2000). This implies that the errors in simulated downward short-wave radiations flux are of the order of 10%. At the same time, the difference between co-albedo (i.e. amount of absorbed shortwave radiation) between clean fresh snow and old dirty snow is 500% (0.1 vs. 0.5).

**Technical corrections**

P9 Fig 6 The location of the two sites are not indicated by black boxes.

The location of the sites are indicated by black boxes, perhaps a bit too small. We will make them more visible.

P9 L15 Replace Nord by Northern

Done.

P9 L16 && L18 Separately instead of "in isolation"

Done.

P11 Fig 9 there is no blue boxes in Fig 5.

Corrected, the blue boxes are shown in Fig. 8a.

Bibliography

Bonelli, S., Charbit, S., Kageyama, M., Woillez, M.-N., Ramstein, G., Dumas, C., and Quiquet, A.: Investigating the evolution of major Northern Hemisphere ice sheets during the last glacial-interglacial cycle, Clim. Past, 5, 329–345, doi:10.5194/cp-5-329-2009, 2009.

Fyke, J. G., Weaver, A. J., Pollard, D., Eby, M., Carter, L., and Mackintosh, A.: A new coupled ice sheet/climate model: description and sensitivity to model physics under Eemian, Last Glacial Maximum, late Holocene and modern climate conditions, Geosci. Model Dev., 4, 117-136, https://doi.org/10.5194/gmd-4-117-2011, 2011.

Helsen, M. M., van de Wal, R. S. W., Reerink, T. J., Bintanja, R., Madsen, M. S., Yang, S., Li, Q., and Zhang, Q.: On the importance of the albedo parameterization for the mass balance of the Greenland ice sheet in EC-Earth, The Cryosphere, 11, 1949-1965, https://doi.org/10.5194/tc-11-1949-2017, 2017.

Musilova M, Tranter M, Bamber JL, Takeuchi N, Anesio AM. Experimental evidence that microbial activity lowers the albedo of glaciers. Geochem Perspect Lett.,doi:10.7185/geochemlet.1611, 2016.

Stibal M, Box JE, Cameron KA, Langen PL, Yallop ML, Mottram RH, et al. Algae drive enhanced darkening of bare ice on the Greenland ice sheet. Geophysical Research Letters, 44, doi:10.1002/2017GL075958, 2017.

**References**

Abe-Ouchi, A., Segawa, T. and Saito, F.: Climatic conditions for modelling the Northern Hemisphere ice sheets throughout the ice age cycle, Clim. Past, 3(3), 423–438, doi:10.5194/cp-3-423-2007, 2007.

Bauer, E. and Ganopolski, A.: Aeolian dust modeling over the past four glacial cycles with CLIMBER-2, Glob. Planet. Change, 74(2), 49–60, doi:10.1016/j.gloplacha.2010.07.009, 2010a.

Bauer, E. and Ganopolski, A.: Aeolian dust modeling over the past four glacial cycles with CLIMBER-2, Glob. Planet. Change, 74(2), 49–60, doi:10.1016/j.gloplacha.2010.07.009, 2010b.

Bauer, E. and Ganopolski, A.: Sensitivity simulations with direct shortwave radiative forcing by aeolian dust during glacial cycles, Clim. Past, 10(4), 1333–1348, doi:10.5194/cp-10-1333-2014, 2014.

Bonelli, S., Charbit, S., Kageyama, M., Woillez, M.-N., Ramstein, G., Dumas, C. and Quiquet, A.: Investigating the evolution of major Northern Hemisphere ice sheets during the last glacial-interglacial cycle, Clim. Past, 5(3), 329–345, doi:10.5194/cp-5-329-2009, 2009.

Calov, R., Ganopolski, A., Claussen, M., Petoukhov, V. and Greve, R.: Transient simulation of the last glacial inception. Part I: glacial inception as a bifurcation in the climate system, Clim. Dyn., 24(6), 545–561, doi:10.1007/s00382-005-0007-6, 2005.

Charbit, S., Ritz, C., Philippon, G., Peyaud, V., Kageyama, M., Charbit, S., Ritz, C., Philippon, G., Peyaud, V. and Numerical, M. K.: Numerical reconstructions of the Northern Hemisphere ice sheets through the last glacial-interglacial cycle To cite this version : of the Past Numerical reconstructions of the Northern Hemisphere ice sheets through the last glacial-interglacial cycle, , 15–37, 2007.

Dickinson, R. E., Henderson-Sellers, A., Kennedy, P. J. and Wilson, M. F.: Biosphere-atmosphere transfer scheme (BATS) for the NCAR Community Climate Model, J. Clim., (December), 72, doi:10.5065/D67W6959, 1986.

Ganopolski, A., Calov, R. and Claussen, M.: Simulation of the last glacial cycle with a coupled climate ice-sheet model of intermediate complexity, Clim. Past, 6(2), 229–244, doi:10.5194/cp-6-229-2010, 2010.

Gregoire, L. J., Payne, A. J. and Valdes, P. J.: Deglacial rapid sea level rises caused by ice-sheet saddle collapses., Nature, 487(7406), 219–22, doi:10.1038/nature11257, 2012.

Kohfeld, K. E. and Harrison, S. P.: DIRTMAP: The geological record of dust, Earth-Science Rev., 54(1–3), 81–114, doi:10.1016/S0012-8252(01)00042-3, 2001.

Liakka, J., Löfverström, M. and Colleoni, F.: The impact of the North American glacial topography on the evolution of the Eurasian ice sheet over the last glacial cycle, Clim. Past, 12(5), 1225–1241, doi:10.5194/cp-12-1225-2016, 2016.

Lunt, D. J., Foster, G. L., Haywood, A. M. and Stone, E. J.: Late Pliocene Greenland glaciation controlled by a decline in atmospheric CO2 levels., Nature, 454(7208), 1102–5, doi:10.1038/nature07223, 2008.

Mahowald, N. M., Muhs, D. R., Levis, S., Rasch, P. J., Yoshioka, M., Zender, C. S. and Luo, C.: Change in atmospheric mineral aerosols in response to climate: Last glacial period, preindustrial, modern, and doubled carbon dioxide climates, J. Geophys. Res. Atmos., 111(10), doi:10.1029/2005JD006653, 2006.

Petoukhov, V., Ganopolski, a., Brovkin, V., Claussen, M., Eliseev, a., Kubatzki, C. and Rahmstorf, S.: CLIMBER-2: a climate system model of intermediate complexity. Part I: model description and performance for present climate, Clim. Dyn., 16(1), 1–17, doi:10.1007/PL00007919, 2000.

Tarasov, L. and Richard Peltier, W.: Greenland glacial history and local geodynamic consequences, Geophys. J. Int., 150(1), 198–229, doi:10.1046/j.1365-246X.2002.01702.x, 2002.

Willeit, M. and Ganopolski, A.: PALADYN v1.0, a comprehensive land surface–vegetation–carbon cycle model of intermediate complexity, Geosci. Model Dev., 9(10), 3817–3857, doi:10.5194/gmd-9-3817-2016, 2016.

Zweck, C. and Huybrechts, P.: Modeling of the northern hemisphere ice sheets during the last glacial cycle and glaciological sensitivity, J. Geophys. Res., 110(D7), D07103, doi:10.1029/2004JD005489, 2005.

---

## Author Comment (AC2) · 12 Feb 2018

**Response to Referee #2**

Willeit and Ganopolski show the importance of considering the effects of snow aging and dust on the snow albedo and consequently on satisfactorily simulating glacial cycles. The article is well written and its relevance is properly justified. In my opininon, the novely of the paper does not lie directly on the results but on the presentation of the parameterizations for accounting on the mentioned effects on snow albedo. Accordingly, the main weakness of the study is reproducibility. The authors should expand on the snow albedo parameterization in order to other groups being able to reproduce (and benefit from) the current study.

We would like to thank the reviewer for his comments on our manuscript. We have responded to the issues raised by the reviewer below. The original reviewer comments are in black, our responses in blue.

**General comments**

About reproducibility:

Ice sheet – climate coupling represents a considerable ongoing effort for modeling groups. The authors of this article have already convincingly shown in previous studies the necessity of accounting for the snow albedo reduction from ice aging and dust in order to successfully simulate a deglaciation. This article furtherly contributes to this idea and presents the needed albedo parameterizations to do so. This later aspect can be of great importance to groups currently starting to couple GCMs to thermodynamical ice sheet models. Thus, these parameterizations need to be accordingly described.

In the revised version of the paper we included a more detailed description of the albedo parameterization in the model.

1. In page 3, line 14, the snow age factor parameterization is described:

1.1 It might be obvious, but the reader could wonder whether the aging of the snow can simply be computed as a function of temperature and snowfall. Please, elaborate on this and add references.

In reality, the aging of snow is a very complicated process and physically based modeling of snow aging effect on albedo requires high resolution multilayer snowpack models forced by realistic synoptic variability of all climate and hydrological components (e.g. Brun et al., 1992; Flanner et al., 2007). And, as intercomparison projects show, even such complex models simulate very different snow albedo. Needless to say that such approach cannot be used in the CLIMBER model. This is why we have no other choice as to use the only two available characteristics – temperature and precipitation – to parameterize the effect of snow aging on surface albedo. These two characteristics exert primary control on snow aging effect because the growth rate of ice crystals depends strongly on surface temperature while frequent precipitations reduces average size of snow crystals at the surface of snow cover. Our parameterization is simple but still captures the first order effect. This is definitely much better than doing nothing because, as our manuscript shows, this effect is vitally important for simulating realistic glacial cycles.

1.2 The definition of T_0 is missing.

Has been included.

1.3 The age factor is used to represent the grain size. And Fig.1 shows grain radius. How is CLIMBER-2 translating each other? It is linear? Please provide the related expression.

The relation between snow grain radius and snow age factor is now included in the paper.

1.4 Fig.1: Besides the pure snow case, CLIMBER-2 seems to be underestimating the albedo compared to the two other parameterizations. Why? A potential explanation is given by the sentence: "... explained by the choice of the imaginary refractive index of dust". Please, be more specific. On the other hand, the effect of the alternative parameterizations on simulating the glacial cycle is described in the Results section, but it s not explained. I imagine this can simply be a matter of "tunning". Re-calibrating the age factor (or other components of the model) for the two alternative approaches will produce a successful ice-volume evolution. If this is the case, please acknowledge in the paper. Otherwise, the reader remains wondering about the realism of the different approaches.

Yes the difference can be mainly explained by the choice of the imaginary refractive index of dust, which, as shown in different studies and as also mentioned in this paper, varies largely as a function of mineral dust composition. One value might be more appropriate for one source of dust and another for a different source. It is therefore problematic to use one constant global value. On the other hand, tracking poorly known properties of dust based on its origin seems challenging.
In the revised version of the manuscript we now mention that a retuning of the model could possibly allow to successfully reproduce glacial cycles even with alternative albedo parameterisations. The message of the paper should not be that a successful simulation of glacial cycles is possible only with the CLIMBER-2 representation of albedo, but that slightly different schemes could result in very different outcomes.

2. In page 10, the effects of considering aeolian and glaciogenic dust individually are discussed. The interactive aeolian dust representation is conveniently described in previous studies. I could not, however, find the equivalent for glaciogenic dust. How is glaciogenic dust generated in CLIMBER-2? Please provide the necessary information. Furthermore, when Fig.7 shows glaciogenic dust as a necessary condition for a full deglaciation.

More details on the processes generating glaciogenic dust in the model are included in the revised version of the paper. Additional details can be found in Ganopolski et al. (2010), Appendix A.

About discussing the necessity of including a dust cycle:

In the Conclusions section it can be read: "In this study we used an Earth system model of intermediate complexity to show that a proper parameterisation of snow albedo over ice sheets is a crucial ingredient for a successful simulation of the last glacial cycle." This and previous studies from these authors support this conclusion. Nevertheless, other models/groups have shown succesfull glacial cycle simulations without the necessity to invoke "a proper parameterisation of snow albedo". For example, in Abe-Ouchi et al 2007 CP and 2013 Nature, the ablation-isostatic adjustment feedback together with elevation and other feedbacks appear to represent enough processes to simulate the deglaciation.

First, the deglaciation simulated by Abe-Ouchi et al. is not so good. Fig 1d in Abe-Ouchi et al (2013) (and this is obviously their best simulation) shows that Northern Hemisphere ice sheets with the volume corresponding to 20 meters in sea level survived the last deglaciation. This is a lot and by our standard such experiment cannot be considered as successful simulation of deglaciation. Second, Abe-Ouchi et al (2007, 2013) used the PDD scheme. This scheme does not account for albedo at all but it can melt a lot of ice when necessary, partly for the wrong reason (see Bauer and Ganopolski (2017)). In that paper we demonstrated that with properly selected PDD parameters, we also are able to simulate a reasonably realistic glacial cycle (see Fig 10 in Bauer and Ganopolski, 2017). This fact, however, tells us nothing about the importance of surface albedo. The later can only be studies with the models which are based on the physically sounded energy balance approach.

The current main conclusion (see above) of this paper give rise to interesting related questions: Could CLIMBER-2 simulate a deglaciation without considering the effects of dust on snow albedo? If affirmative, which are then the key processes? Are those other processes equally realistic? Is all the relevant physics necessary for understanding deglaciations already contained in EMICs? ... I understand that the authors could see these questions as out of the scope for the current article, but I also believe the readers will appreciate further the current paper if a discussion on this aspect is included.

We here repeat what we have already written in response to one of the comments of reviewer #1. The fact that CLIMBER-2 is capable of reproducing glacial-interglacial cycles is the result of many years of work, which included model improvements, inclusion of new processes based on new process understanding and obviously also tuning of unconstrained model parameters. Even with a relatively fast model like CLIMBER-2 it is practically impossible to explore the whole parameter space and it could well be that other parameter combinations could lead to a successful simulation of glacial cycles. This is beyond the scope of the present study. The goal of this study is to clearly show that surface albedo plays a crucial role for ice sheet evolution and not only do fundamental processes affecting snow albedo, such as snow aging and dust deposition, play an important role, but even using slightly different parameterisations of the same process (e.g. the effect of mineral dust impurities on snow albedo) can lead to qualitatively very different results. This is a clear message to readers who are interested in simulating glacial cycles and more in general the long-term evolution of ice sheets.
We will include a bit more discussion on this aspect in the revised version of the paper.

**Specif comments:**

Page 1, line 10 and 14: Please use "light-absorbing ..." as later in the paper.

Changed.

Page 3, line 8: add "in" afer "snow albedo used..."

Done.

Caption figure 9: erratum: glaciogenic

Fixed.

**References**

Bauer, E. and Ganopolski, A.: Comparison of surface mass balance of ice sheets simulated by positive-degree-day method and energy balance approach, , 819–832, 2017.

Brun, E., David, P., Sudul, M. and Brunot, G.: A numerical model to simulate snow-cover stratigraphy for operational avalanche forecasting, J. Glaciol., 38(128), 13–22, doi:10.1017/S0022143000009552, 1992.

Flanner, M. G., Zender, C. S., Randerson, J. T. and Rasch, P. J.: Present-day climate forcing and response from black carbon in snow, J. Geophys. Res. Atmos., 112(11), 1–17, doi:10.1029/2006JD008003, 2007.